## Research Article

Floods; El Niño-Southern Oscillation; Peru; Low and Middle Income Countries; Mental Health; Child; Adolescent; Matched-Pair Analysis

**Corresponding author:**
Ilan Cerna-Turoff;
Email: it2208@caa.columbia.edu

# El Niño-driven flooding and mental health symptomology among adolescents and young adults in Peru

Ilan Cerna-Turoff[1] , Hyunseung Kang[2] and Katherine M. Keyes[3]

[1]Department of Environmental Health Sciences, Mailman School of Public Health, Columbia University, New York City, NY, USA; [2]Department of Statistics, University of Wisconsin-Madison, Madison, WI, USA and [3]Department of Epidemiology, Mailman School of Public Health, Columbia University, New York City, NY, USA

## Abstract

Intensifying storms and inter-annual El Niño events may increase psychological stress and worsen mental health. This study examines the relationship between flood exposure and long-term mental health symptoms among adolescents and young people in Peru, the world's most affected country by El Niño. We analyzed community and self-reported survey data from the Young Lives Study to contrast mental health in 2016 among youth who lived in communities that experienced or did not experience flooding between 2013 and 2016. We pre-processed data on 1344 individuals in 93 communities, using optimal full matching on Mahalanobis distance with a propensity score caliper, and estimated relative risks to mental health scores in the general population of young people and among gender-stratified groups via quasi-Poisson regression. Exposure to floods did not yield conclusive differences in mental health scores in this sample. Further evidence is needed on mental health patterns over time, the influence of exposure severity, and the impact of disaster relief on symptomology in mounting an effective global health response.

## Impact statement

Extreme weather frequently and disproportionately affects low- and middle-income countries. Information on the effect of natural hazard exposure on mental health is limited outside of high-income countries and is lacking in Latin America. When existent, the literature is often restricted to adults, despite young people comprising the majority of the population in low- and middle-income countries. This study provides high-quality evidence to expand each of these evidence gaps. Mental health was not conclusively different among exposed young people in the years after flooding in this sample, which implies that interventions in this population may need to be delivered in a shorter window of time after exposure. Our research is further a renewed call for mental health research in a greater variety of contexts and populations globally. This study contributes to the body of evidence on mental health among young people in Peru, the country most affected by El Niño and gender-specific patterns of mental health. In addition, climate change is propelling the need for evidence from low- and middle-income countries. Governments and humanitarian response need robust, contextualized evidence to mount effective and equitable mental health services after climate-related disaster events.

## Introduction

Over the last four decades, we have experienced a global intensification of extreme weather and a regional and seasonal shift in meteorological patterns. Heavy precipitation has likely increased globally and on the continental scale, driven by surges in precipitation over North America, Europe, and Asia (Intergovernmental Panel on Climate Change, 2023). Across the equatorial Pacific, the world's largest mode of climate variability, known as "El Niño Southern Oscillation," usually consists of inter-annual cycles of rainfall and dry conditions produced by feedback loops between wind strength and ocean temperatures (Sanabria et al., 2018). Unprecedented rises in temperatures have yielded longer and more frequent fluctuations in air pressure, increasing the likelihood of heavy rainfall, drought, and severe storms (Kovats et al., 2003; Wang et al., 2019). Peru is the world's most affected nation by El Niño, and while the country experiences an annual rainy season, heavy rainfall and flooding often occur during El Niño years (French and Mechler, 2017; Sanabria et al., 2018).

Extreme precipitation can have tremendous social, economic, and environmental implications. Particularly in low- and middle-income countries, 19% of all damage and loss to agriculture is directly traced to flooding and an additional 18% to extreme storms. Between 2008 and 2018,

floods account for 21 billion USD in losses globally. These losses mark floods as the second costliest type of disaster after drought (FAO, 2021). Within the health domain, flooding affects physical health via several pathways. The most direct is mortality from drowning, followed by mortality and injury from infrastructure collapse or blockage and medical service inaccessibility. Water contamination and water-borne pathogens are often secondary consequences of floods, as sewer, drainage, and other containment systems are overwhelmed, and water stands for extended periods. Mold and booms of mosquito populations further lead to poor respiratory health and transmission of communicable diseases like dengue fever (Ahern et al., 2005; Coalson et al., 2021; Doocy et al., 2013).

A conglomeration of factors increases psychological stress and mental health conditions after floods. Flooding can lead to the loss of family members, homes, and other possessions. Flooding may additionally result in displacement to areas where households have decreased safety and social networks. Initial loss and new contextual risks combine to increase the likelihood of negative mental health (Fussell and Lowe, 2014). Households that remain in flood-affected zones face cumulative stressors related to rebuilding after economic loss, particularly among the poorest households (Berry et al., 2010; Stanke et al., 2012). A meta-analysis of post-traumatic stress disorder (PTSD) among 40,600 individuals found that PTSD was higher among those who had experienced moderate versus mild floods and that the first six months after floods had a higher incidence than subsequent time periods (Chen and Liu, 2015). Several other studies have found that flooding worsened mental health symptoms and conditions in China (Liu et al., 2006), the United States (Ginexi et al., 2000), Australia (Bei et al., 2013), and the United Kingdom (Waite et al., 2017). Less is known about mental health outside of these isolated geographic areas. Given the focus of epidemiological study on PTSD, most evidence does not capture other domains of mental health that may extend over longer time periods (Chen and Liu, 2015; Stanke et al., 2012). In addition, studies tend to focus on adult populations. The body of evidence on adolescents and young adults is limited to a few isolated studies in which participants exhibited less anxiety and depression symptoms than their older counterparts (Liu et al., 2006; Peng et al., 2011). This study examines how community exposure to floods affected mental health symptomology among a cohort of adolescents and young adults in Peru. We sought to quantify potential differences in mental health scores among those exposed and unexposed to flooding, primarily driven by El Niño, and to identify gendered differences in stratified analyses.

## Methods

**Data collection and measurement**. We analyzed data from the Young Lives Study, a longitudinal study of 2766 young people in Peru who were followed from birth to young adulthood. Young Lives is one of the largest and oldest cohort studies of childhood poverty outside of high-income countries and captures an array of demographic, health, and economic information. The study was gender balanced and used a three-stage random sampling design of children within households ("*Young Lives methods guide: The longitudinal survey*," 2011). The sample population had relatively similar characteristics to past nationally representative surveys from Peru, with households having slightly more assets and access to basic services than those in the 2000 Demographic and Health Survey (DHS), 2001 Peru Living Standard Measurement Survey

(LSMS), and 2005 national census (Escobal and Flores, 2008). The research team administered surveys in three-to-four-year intervals with adolescents and young adults, community respondents, and members of young people's households ("*Young Lives methods guide: The longitudinal survey*," 2011). We restricted our analysis to data collection in 2013 and 2016. The final sample consisted of adolescents and young adults who were 14-to-15 and 20-to-25 years of age, respectively, in 2016 (Boyden et al., 2018a; Boyden et al., 2018b).

**Flood exposure**. Retrospective information on flood exposures was collected in 2016. Social service providers and community leaders reported on flooding in the past three years (2013 to 2016) in districts where young people lived (Figure 1). We coded exposure as a binary of exposed if a community was reported to have experienced one or more flood events in the past three years and unexposed if no natural hazard was reported. This reporting period had low levels of precipitation; El Niño was the strongest driver of flooding, overlapping with the reporting period in 2015 and 2016 (Sanabria et al., 2018). Communities were excluded if they were exposed to another natural hazard during the reporting time period to isolate potential differences in mental health related to flooding alone. Since repeat surveys were collected with young people, and past migration history was reported, we were able to identify where participants lived to assign exposure and if participants migrated to different communities between 2013 and 2016. We restricted to individuals who remained in the same communities to reduce potential biases in misclassification of exposure and to avoid the introduction of additional, unknown community-level factors that may have confounded analyses.

**Mental health outcome**. Adolescents and young adults were directly asked a series of five questions in 2016 related to anxiety, depression, and possible psychosomatic symptoms: (1) worry a lot, (2) have a lot of headaches, stomach aches, or sickness, (3) are often unhappy, downhearted, or tearful, (4) are nervous in new situations, and (5) have many fears or are easily scared. The response options were "not true", "a little true", or "certainly true". We assigned a value of 0 to 2 to each question to create scores that ranged from 0 to 10, with higher numbers indicating more symptoms. These questions overlap with two psychometrically validated screeners of anxiety and depression in the general population—the Generalized Anxiety Disorder-7 (GAD-7) and the Patient Health Questionnaire depression-8 (PHQ-8)—and include an additional question on general physical well-being not found in the screeners (Kroenke et al., 2009; Spitzer et al., 2006). The GAD-7 and PHQ-8 have been validated in Peru (Carroll et al., 2020; Mughal et al., 2020), and Peruvian populations exhibit psychosomatic symptoms of negative mental health, which merits including physical health as part of a mental health screener (Pedersen et al., 2008). We assessed reliability among items in the newly created scales, using Cronbach's alpha ($\alpha$).

**Covariates**. As previously noted, prior analyses indicated that the participants' baseline characteristics were roughly similar to other nationally representative surveys. We adjusted for individual-, household/caregiver-, and community-level covariates plausible in confounding the relationship between flood exposure and mental health outcomes and available in these data (refer to Supplementary Materials 1). These covariates included: gender, age, race, tobacco use, and past well-being scores on the individual level; household size and caregiver home ownership, job loss, divorce, and education level; and access to a social worker in the community. Peru, as in many low- and middle-income countries, has severely limited mental health services. 85% of psychiatric hospitals were

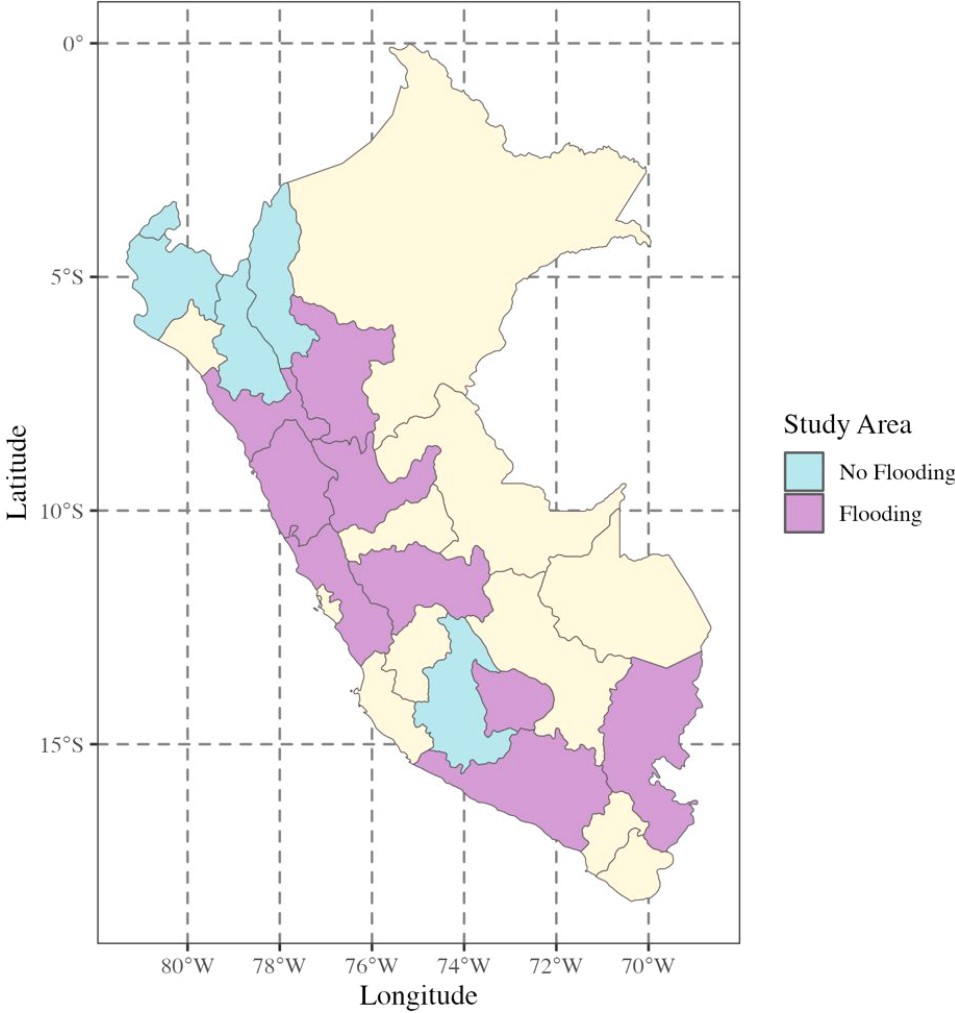

**Figure 1.** EM-DAT data used to indicate administrative areas where flood incidents occurred between 2013 and 2016 (Guha-Sapir et al., n.d.). Administrative areas that include communities where the Young Lives Study was conducted are shaded in blue and purple (Escobal and Flores 2008).

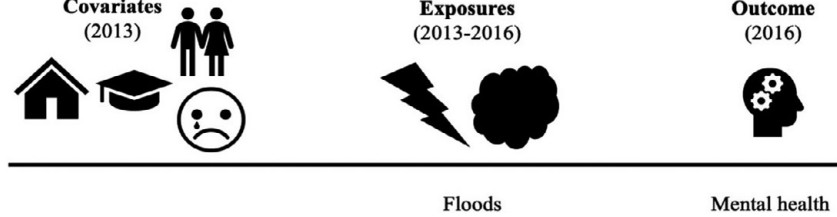

**Figure 2.** Covariates were reported prior to exposures, and outcomes were measured subsequent to exposures.

concentrated in the capital city of Lima, and the first reforms to implement national mental health services as part of primary care began in 2013 (Toyama et al., 2017). The metric of a social worker in communities is therefore a proxy for the availability of social services that may influence mental health during the study time period. Demographic and behavioral factors were constructed from self and household surveys in 2013. All covariates were measured prior to reported community exposures (Figure 2).

**Analysis**. We targeted causal effects in this analysis to understand the average treatment effect on the treated (ATT), which is the potential difference in average mental health scores for young

adults who lived in communities where flooding had occurred had they never been exposed to flooding. We restricted our analysis to individuals who had both exposure and outcome, which constituted approximately 62.1% of the total population ($n = 1718$). The sample was further restricted to areas that did not experience other types of natural hazards ($n = 1348$) and excluded clusters that had one observation after applying the exclusion criteria ($n = 1344$). 25 individuals moved communities between 2013 and 2016 but were excluded in applying the other exclusion criteria ($n = 1344$) (refer to Supplementary Materials 2). Given missingness in covariates (< 1.0 to 77.2%), we ran a series of five multiple imputations in

**Table 1.** Sample characteristics of complete cases in areas affected by flooding and no flooding between 2013 and 2016

| Variable | Overall | No flooding | Flooding |
|---|---|---|---|
| | (n = 1344) | (n = 1000) | (n = 344) |
| **Gender**, n (%) | | | |
| Female | 648 (48.2%) | 488 (48.8%) | 160 (46.5%) |
| Male | 696 (51.8%) | 512 (51.2%) | 184 (53.5%) |
| **Age**, mean (SD) | 13.08 (3.00) | 13.15 (3.05) | 12.87 (2.83) |
| **Race**, n (%) | | | |
| Mestizo/a | 1,221 (90.9%) | 897 (89.7%) | 324 (94.5%) |
| White | 81 (6.0%) | 70 (7.0%) | 11 (3.2%) |
| Other | 41 (3.1%) | 33 (3.3%) | 8 (2.3%) |
| **Tobacco use**, n (%) | | | |
| Yes | 138 (45.0%) | 108 (45.2%) | 30 (44.1%) |
| No | 169 (55.0%) | 131 (54.8%) | 38 (55.9%) |
| **Well-being score**, mean (SD) | 6.47 (1.79) | 6.36 (1.77) | 6.79 (1.81) |
| **Household size**, mean (SD) | 5.09 (1.84) | 5.10 (1.86) | 5.08 (1.78) |
| **Caregiver home ownership**, n (%) | | | |
| Yes | 1,028 (76.7%) | 736 (73.8%) | 292 (84.9%) |
| No | 313 (23.3%) | 261 (26.2%) | 52 (15.1%) |
| **Caregiver job loss**, n (%) | | | |
| Yes | 91 (6.8%) | 70 (7.0%) | 21 (6.1%) |
| No | 1,253 (93.2%) | 930 (93.0%) | 323 (93.9%) |
| **Caregiver divorce**, n (%) | | | |
| Yes | 73 (5.4%) | 54 (5.4%) | 19 (5.5%) |
| No | 1,271 (94.6%) | 946 (94.6%) | 325 (94.5%) |
| **Caregiver education level**, n (%) | | | |
| None | 75 (7.3%) | 59 (7.9%) | 16 (5.8%) |
| Basic | 328 (32.1%) | 219 (29.3%) | 109 (39.6%) |
| Above Basic | 620 (60.6%) | 470 (62.8%) | 150 (54.5%) |
| **Social worker in community**, n (%) | | | |
| Yes | 719 (53.5%) | 556 (55.6%) | 163 (47.4%) |
| No | 625 (46.5%) | 444 (44.4%) | 181 (52.6%) |

SD: standard deviation. The study participants were adolescents and young adults who were 14-to-15 and 20-to-25 years of age, respectively, in 2016. Sample sizes that are less than the overall total in individual covariates indicate missingness and were imputed in the final analysis. Continuous variables are measured as mean values and standard deviations.

the `Mice` package in the R statistical software (Van Buuren and Groothuis-Oudshoorn, 2011). Outcomes were not included in the imputation to avoid inducing bias (Lee et al., 2014). We applied the `MatchThem` package to create sets of exposed and unexposed observations within the five imputations and pooled them for analysis (Pishgar et al., 2021). Love plots were utilized to assess adequate bias reduction of matching algorithms on the imputed datasets, and a standardized mean difference of 0.1 or below was deemed the threshold for sufficient balance (refer to Supplementary Materials 3) (Rubin, 2001). We selected optimal full matching on Mahalanobis distance with a propensity score caliper to achieve balance (Hansen, 2004). We restricted the algorithm so that each exposed unit was matched to between one to three unexposed units, omitting up to 60% of the reservoir of unexposed observations across imputations. We combined a series of five questions in the

survey that related to mental health symptoms to create a score, ranging from zero to ten, to use as the outcome. We excluded individuals who were missing three or more questions in the series (4.4%) and conducted multiple imputations across the five questions for calculating a complete mental health outcome score. These questions utilized for outcome score creation exhibited high levels of internal consistency (α = 0.82, 95% confidence intervals [CI]: 0.79–0.85). We fitted quasi-Poisson regression with a log link, which accommodates the slight overdispersion of outcomes. We used the recommended sampling weights from `MatchThem` in the `Survey` package to obtain standard errors after matching (Pishgar et al., 2021). Effects were measured as relative risks. In addition, we ran gender-stratified models to identify potential differences for young women and men separately. With observational data, it is never certain that covariate adjustment is sufficient to remove all

sources of confounding. We therefore conducted sensitivity analyses for robustness to unmeasured confounding, using the `EValue` package and *E*-value calculator (Mathur et al., 2018). *E*-values suggest robustness to unmeasured confounding by indicating the minimum strength of association that an unmeasured confounder would need to have with the exposure and outcome to move the observed point estimate to a different specified effect on the risk ratio scale, above and beyond the measured covariates (VanderWeele and Ding, 2017). In this analysis, we sought to identify what amount of unmeasured confounding would move risk ratios from the estimated value to a "true" relative risk of two, which was deemed a strong indicator that flooding meaningfully impaired mental health. All data preparation and statistical analyses were conducted with R statistical software, version 4.1.2 (Cerna-Turoff et al., 2023; R Core Team, 2021).

## Results

**Descriptive statistics of the study population.** The final sample consisted of 1344 individuals in 93 communities. Between 2013 and 2016, a total of 344 individuals were exposed to flooding, and 1000 were not exposed. The number of young women and men was roughly equivalent (648 women and 696 men). Most young people identified as mixed race ("mestizo/a"), followed by small percentages who identified as "White" or "Other" (i.e., Asian, Black, or Indigenous). Nearly half of young people recently used tobacco products. The sampled households exhibited mixed indicators in terms of their overall poverty levels (for instance, over 76.7% owned their own home, indicating some degree of wealth, but the respondents' caregivers also had low levels of education, which is often correlated with poverty) (Filmer and Pritchett, 1999). Over half of the study communities did not have access to social services. These characteristics were roughly comparable between areas affected by flooding and non-flooding (Table 1).

Within the imputed sample, approximately one-third of young people lived in communities that were exposed to flooding at some point during the study period (34.4%). Prior to exposure, the average well-being scores of young people in the sample were relatively high (6.48 out of 10), and women had slightly better well-being scores than men (6.55 and 6.41, respectively). After exposure, most young adults had low mental health scores, with an average score of 3.01.

**Mental Health Symptoms**. Flooding did not have a clear directional effect on mental health scores in the unadjusted sample. Mental health scores among young people were slightly lower when they lived in communities that experienced flooding compared to communities that did not experience flooding (relative risks [RR] = 0.99, 95% CI: 0.96–1.03). The relative risks were likewise lower among exposed women (RR = 0.97, 95% CI: 0.93–1.01) and slightly higher among

**Table 2.** Effect of flooding on mental health symptoms

|  | Relative risks | 95% Confidence intervals | *p*-values |
|---|---|---|---|
| General | 1.02 | 0.90–1.16 | 0.73 |
| Women | 0.97 | 0.850–1.11 | 0.65 |
| Men | 1.06 | 0.880–1.28 | 0.52 |

Rounded to two decimal places. Scale for mental health created from a series of five questions in the survey related to anxiety, depression, and possible psychosomatic symptoms. Adjusted by factors related to the individual respondent (gender, age, race, tobacco use, and past well-being score), household or caregiver-related factors (household size, home ownership, recent job loss, recent divorce, education level), and presence of social workers in the community.

exposed men (RR = 1.02, 95% CI: 0.97–1.07). In the adjusted sample, we did not find evidence of significantly higher mental health scores among young people who lived in communities where flooding occurred. The findings were likewise inconclusive as to flooding increasing or decreasing mental health symptoms among young women and men (Table 2).

Sensitivity analyses indicated that moderate amounts of unmeasured confounding would be needed to move the current relative risk to two for the full sample (*E*-value = 3.33). Similarly, gender-stratified results would require a substantive amount of unmeasured confounding to shift the point estimate to a relative risk of two (*E*-value women = 3.54; *E*-value men = 3.18) (refer to Supplementary Materials 4).

## Discussion

We sought to expand the empirical evidence on the relationship between flooding and mental health symptoms among adolescents and young adults in Peru within a stratified sample by gender. In this analysis, we found that flooding did not have a clear effect on mental health symptoms and that a moderate amount of unmeasured confounding would be needed to shift the observed results to a larger and more meaningful effect size. Past literature has indicated that children and adolescents tend to exhibit better mental health after exposure to floods than other age groups (Liu et al., 2006; Peng et al., 2011). It may be that this sample experienced non-severe mental health symptomology overall or that their mental health equilibrated relatively rapidly. Other contextual factors may likewise have exerted an influence. An important element of exposure is flood severity. Unlike previous El Niño years, Peru experienced uncharacteristically low levels of precipitation from 2015 to 2016 (Sanabria et al., 2018). A certain threshold may exist at which flooding leads to negative mental health symptoms, which resonates with existing evidence (Chen and Liu, 2015; Liu et al., 2006). In addition, between 2015 and 2016, the Peruvian government invested $1.53 billion USD in disaster preparedness (World Food Programme, 2016). The large-scale investment may have buffered the effects of flooding (Simonds et al., 2022), providing valuable resources to disaster-prone areas of the country. While an evaluation study indicated that funding was not effectively distributed at local levels of administration (French and Mechler, 2017), it still represents a massive and unprecedented national investment. This influx of funds contrasts with access to social services. Only half of community respondents reported that their communities had access to social workers who could provide mental health services. It is unclear how the particular mix of investment and lack of service providers impacted population-level mental health. A shorter window of time also may exist for identifying the signal between flooding and negative mental health symptoms. We categorized communities as exposed if they experienced flooding between 2013 and 2016. The effects may not have been sustained over the three-year time period. Other studies of mental health indicate a gradual recovery for the majority of the affected population. Adolescent survivors of a major earthquake in China, for instance, exhibited elevated depressive symptoms that diminished after one and a half years (Ye et al., 2014). Studies after disaster events in contexts as diverse as Nepal, Thailand, and New Zealand show persistent depression and anxiety symptomology in the affected population over extended but variable time periods of several years (Beaglehole et al., 2022; Silwal et al., 2022; van Griensven et al., 2006). Further investigation is needed on the

mental health trajectory of adolescents and young adults to identify whether a key window of vulnerability exists.

**Limitations.** We applied a rigorous suite of methods to estimate the effect of flood exposure on mental health in a dataset that was largely representative of the population of Peru. We isolated communities that were exposed to flooding versus no other form of natural hazard and exploited the randomness and extreme variability in heavy precipitation on the local scale to conduct a causal analysis (Sanabria et al., 2018). A main limitation is that we may have not removed all residual sources of confounding. Observational studies often do not measure or have access to all covariates that bias estimates. In particular, we did not have access to detailed metrics on the types of disaster relief activities conducted in communities. In addition, our results may be sensitive to how poverty was measured. Poverty is an important stressor that can heighten the negative effect of extreme weather events (Bonanno et al., 2007; Cerda et al., 2010). We included property ownership in covariate adjustment, but poverty is a multi-dimensional phenomenon. No standardized metric for all dimensions of poverty exists, especially in low- and middle-income countries. Proxy measures range from measuring the construction material of houses and assets to consumption and income directly (Oscar Rutstein and Johnson, 2004). Furthermore, sensitivity analyses indicated that moderate amounts of residual confounding would have been needed to increase our point estimate if the "true" effect was a risk ratio of two. Relatedly, communities frequently experienced more than one natural hazard type between 2013 to 2016 time period, which we would expect to worsen mental health and bias away from the null. This is analogous to other competing exposures in observational studies. We did not, however, have a strong directional effect on mental health symptoms in the exposed population, indicating that overlapping natural hazard exposures did not likely alter our results. Another limitation is that the mental health scores were not measured at multiple points, which hinders testing of the effect of flood exposure within different windows of time. Bias could also have been introduced as part of covariate imputation. A few covariates with low levels of missingness exhibited wide confidence intervals across imputed datasets in the unadjusted sample (refer to Supplementary Materials 3), suggesting that extreme values may bias other covariates during imputation. If covariates are not predictive of the outcome or treatment, biases that may have arisen during imputation would not be highly concerning. However, some covariates, such as gender or prior low scores for well-being, have been found to be strongly associated with future mental health (Norris et al., 2002). We intensively compared different matching algorithms to select a method that balanced these key covariates so that the range of covariate values remained within 0.1 SMD across the five imputations. It does not, however, negate that standard errors will increase outside of these bounds as the number of imputations increases. We further have the possibility of measurement error in the exposure. In this sample, community leaders reported flood exposure. It is possible that some communities may have been misclassified as exposed or not reported. Apart from direct measures of soil moisture, precipitation, or another physical test of water and climatic conditions, it is presumable that community leaders would know the conditions of their communities, particularly for sudden surges in the river or coastal zones or heavy precipitation. Measurement errors may also have occurred in the outcome. Outcome scores exhibited good internal consistency, captured information on symptoms that are characteristic of anxiety and depression, and had low levels of missingness in items. Nevertheless, it is likely that they did not capture all dimensions of mental health, given the lack of psychometric testing of the scale. Likewise,

measurement errors may have occurred inadvertently in selecting the comparison group during the matching process. Unexposed units may be selected from across several different communities to be matched to flood-exposed observations, introducing the possibility that several versions of the "placebo" of no flood exposure were combined. We attempted to make each group as homogenous as possible and far apart in their covariate values by excluding communities that were exposed to any other natural hazard. The possibility of measurement error, nonetheless, remains as an artifact of the process of selecting matches.

The changing nature of climate globally is a major issue of our time. As extreme weather intensifies, we have a pressing need to understand the implications for short- and long-term mental health. Past research has largely focused on physical health (Ahern et al., 2005; Coalson et al., 2021; Doocy et al., 2013); however, mental health intersects with multiple domains of life and is integral to individual well-being and public health (GBD 2019 Mental Disorders Collaborators, 2022). The economic costs of mental health are high, with an estimated loss of $5 trillion USD for mental health disorders in 2019 alone (Arias et al., 2022). A better understanding of how extreme weather events impact mental health can yield better allocation of resources and reduction of healthcare costs. Particularly in countries that are highly affected by climate change and El Niño like Peru and where healthcare systems are already stretched, targeted interventions and earmarking of funding have the potential to mitigate mental health burdens on the population level. Moreover, critical gaps exist in our understanding of the post-exposure trajectory of mental health. This study provides new information that the worst effects may not be sustained, which highlights the likely importance of prevention and acute-phase mental health services. In general, we need additional research that applies high-quality methods to isolate the effect of environmental exposures on mental health. A limited number of studies have targeted causal effects after geophysical disasters and to a lesser extent, extreme weather events (Cerna-Turoff et al., 2020; Obradovich et al., 2018; Wertis et al., 2023; Zubizarreta et al., 2013). We present high-quality evidence that should serve as a blueprint for other researchers seeking to understand mental health after extreme weather events and climate-related disasters.

We specifically call on future research to fill gaps in the evidence by quantifying the mental health consequences of extreme weather using rigorous study designs. Data should be collected with sufficient detail on social services, infrastructure, disaster preparedness, and other individual and community characteristics, which may influence the impact of extreme weather on mental health. Analyses can use this information to adjust for systematic differences among communities but also to identify which structures and programs are most protective after extreme weather. Further metrics that measure exposure could likewise be strengthened by examining differences between self- and community-report and climatological measures of devastation on the local level. A composite measure that compiles various dimensions of exposure on a continuous scale supports the identification of which components most impact mental health and how slight changes may lead to differential outcomes. In addition, our epidemiological evidence can better inform disaster response and subsequent investment in social services and public health if we better understand mental health trajectories. Future studies can measure mental health at multiple time points after exposure, including immediately after exposure. Greater measurement of both short- and long-term changes will provide a detailed picture of which populations are impacted immediately and over time to guide investment. Implementation science should parallel epidemiological research. As research

quantifies protective elements of the environment and segments of the population that are worst affected, we can test how the structure of disaster preparedness promotes positive mental health symptoms and create studies comparing the effectiveness of interventions that respond to extreme weather events. Overall, regions and populations at the highest risk of negative mental health and exposure to extreme weather should be the center point of the research agenda. We highlight the need for disaggregation by age and gender and underscore the future need for research that examines differences in social support, socioeconomic status, and other pertinent sociodemographic characteristics. We need to build the evidence base among vulnerable populations, particularly in low- and middle-income countries disproportionately affected by climate change.

## Conclusions

Extreme weather events present a major risk to population health that will continue to intensify in future years. In Peru, we found that exposure to flooding did not increase mental health symptoms among adolescents and young adults or gender-stratified groups. The findings expand our knowledge of mental health among young people in a setting highly affected by extreme weather. We have an urgent need to bridge gaps in our knowledge of populations disproportionately affected by extreme weather in order to mount an effective mental health response. Specifically, we need to build consensus on how to measure exposure, understand mental health trajectories over the short and long term, and to identify groups most at risk of negative mental health symptoms and psychopathology outside of high-income settings. Epidemiological evidence would benefit from linkages to implementation science. Understanding the types of disaster preparedness and response and kinds of interventions that buffer negative mental health symptoms in a broad range of populations is essential to young people's health and well-being, particularly among communities highly affected by climate change and extreme weather events. With the combination of better measurement and evidence-based solutions, we can mount an effective public health response.

**Open peer review.** To view the open peer review materials for this article, please visit http://doi.org/10.1017/gmh.2024.121.

**Data availability statement.** The raw data and study documentation can be accessed from the UK Data Service (https://beta.ukdataservice.ac.uk/datacatalogue/series/series?id=2000060). The code for data analysis can be found on GitHub (https://github.com/ilan-cerna-turoff/code-for-papers).

**Acknowledgments.** We would like to extend our sincere gratitude to Lawrence G. Chillrud for creating automated functions for standardizing the processing and cleaning of the data.

**Author contribution.** Ilan Cerna Turoff: Conceptualization, methodology, software, data curation, formal analysis, writing – original draft, writing – review & editing, visualization, project administration. Hyunseung Kang: methodology, software, writing – review & editing. Katherine Keyes: conceptualization, writing – review & editing.

**Financial support.** This work was supported by the National Institute of Environmental Health Sciences of the National Institutes of Health (grant number: T32ES007322). The content is solely the responsibility of the authors and does not necessarily represent the official views of the National Institutes of Health. The funder had no role in the design, analysis, or other aspects of this publication.

**Competing interest.** None.

**Ethics statement.** Prior to data collection, the study was approved by the Institutional Review Board at the University of Oxford as well as country-specific ethical review boards [24]. This secondary data analysis was deemed exempt from Columbia University's Institutional Review Board (IRB-AAAT932) and performed in accordance with the ethical standards outlined in the 1964 Declaration of Helsinki and later amendments. Informed consent was obtained from all individual participants included in the study.

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
