## [Reviewer Report]

The manuscript is well written and well thought through. The methodology is sound, and the discussion is good. Thank you for including a section about limitations of the study.

---

## [Reviewer Report]

Positive Aspects

The study has addressed a critical and timely issue, considering the increasing frequency of El Niño events and their profound impacts on communities, especially in developing countries like Peru. It has filled a significant gap in the literature concerning the mental health impacts of such environmental disasters on adolescents and young adults in Peru.

The use of optimal full matching and quasi-Poisson regression models has demonstrated a robust analytical approach. This methodological rigour has enhanced the credibility of the findings and has provided a solid foundation for understanding the complex relationships between environmental disasters and mental health.

The analysis of data from 1344 individuals across 93 communities has offered a comprehensive overview of the situation in Peru. The gender-stratified analysis has further enriched the findings by highlighting potential differences in mental health impacts between young men and women.

Areas for Improvement

While the study makes a significant contribution to understanding the mental health impacts of El Niño-driven flooding, it should also open more avenues for further research. The manuscript should provide more explicit recommendations for future studies, such as exploring the long-term psychological effects, the role of disaster preparedness and response, and the effectiveness of different mental health interventions.

In table 1, the age range of the participants where not included. That should be corrected.

In table 1, the wellbeing scores and the household sizes of the participants were not included. That should be corrected.

Conclusion

The manuscript has presented a valuable study on the mental health impacts of El Niño-driven flooding among adolescents and young adults in Peru. The methodological rigour and comprehensive data analysis are commendable. However, further elaboration on recommendations for future research could enhance its contribution to the field.

---

## [Editor Report]

The team of reviewers and myself have been very satisfied with the article, we only have some minor observations that we ask you to clarify. We are sending you the comments below:

Reviewer 1: 

The manuscript is well written and well thought through. The methodology is sound, and the discussion is good. Thank you for including a section about limitations of the study. 

Reviewer 2:

Positive Aspects: The study has addressed a critical and timely issue, considering the increasing frequency of El Niño events and their profound impacts on communities, especially in developing countries like Peru. It has filled a significant gap in the literature concerning the mental health impacts of such environmental disasters on adolescents and young adults in Peru.

The use of optimal full matching and quasi-Poisson regression models has demonstrated a robust analytical approach. This methodological rigour has enhanced the credibility of the findings and has provided a solid foundation for understanding the complex relationships between environmental disasters and mental health.

The analysis of data from 1344 individuals across 93 communities has offered a comprehensive overview of the situation in Peru. The gender-stratified analysis has further enriched the findings by highlighting potential differences in mental health impacts between young men and women.

Areas for Improvement: While the study makes a significant contribution to understanding the mental health impacts of El Niño-driven flooding, it should also open more avenues for further research. The manuscript should provide more explicit recommendations for future studies, such as exploring the long-term psychological effects, the role of disaster preparedness and response, and the effectiveness of different mental health interventions.

In table 1, the age range of the participants where not included. That should be corrected.

In table 1, the wellbeing scores and the household sizes of the participants were not included. That should be corrected.

Conclusion: The manuscript has presented a valuable study on the mental health impacts of El Niño-driven flooding among adolescents and young adults in Peru. The methodological rigour and comprehensive data analysis are commendable. However, further elaboration on recommendations for future research could enhance its contribution to the field.